# Enhancing Mixing Performance in a Rotating Disk Mixing Chamber: A Quantitative Investigation of the Effect of Euler and Coriolis Forces

**DOI:** 10.3390/mi13081218

**Published:** 2022-07-29

**Authors:** Jihyeong Lee, Saebom Lee, Minki Lee, Ritesh Prakash, Hyejeong Kim, Gyoujin Cho, Jinkee Lee

**Affiliations:** 1School of Mechanical Engineering, Sungkyunkwan University, Suwon 16419, Korea; ljh2362@g.skku.edu (J.L.); leesb@skku.edu (S.L.); 2Biomedical Research Institute, Korea Institute of Science and Technology, Seoul 02792, Korea; dlalsrl11@kist.re.kr; 3Institute of Quantum Biophysics, Sungkyunkwan University, Suwon 16419, Korea; iit.ritesh@gmail.com; 4Research Engineering Center for R2R Printed Flexible, Sungkyunkwan University, Suwon 16419, Korea; 5School of Mechanical Engineering, Korea University, Seoul 02841, Korea; h_kim@korea.ac.kr; 6Department of Biophysics, Sungkyunkwan University, Suwon 16419, Korea

**Keywords:** Euler force, Coriolis force, mixing, microfluidics, rotating disk

## Abstract

Lab-on-a-CD (LOCD) is gaining importance as a diagnostic platform due to being low-cost, easy-to-use, and portable. During LOCD usage, mixing and reaction are two processes that play an essential role in biochemical applications such as point-of-care diagnosis. In this paper, we numerically and experimentally investigate the effects of the Coriolis and Euler forces in the mixing chamber during the acceleration and deceleration of a rotating disk. The mixing performance is investigated under various conditions that have not been reported, such as rotational condition, chamber aspect ratio at a constant volume, and obstacle arrangement in the chamber. During disk acceleration and deceleration, the Euler force difference in the radial direction causes rotating flows, while the Coriolis force induces perpendicular vortices. Increasing the maximum rotational velocity improves the maximum rotational displacement, resulting in better mixing performance. A longer rotational period increases the interfacial area between solutions and enhances mixing. Mixing performance also improves when there is a substantial difference between Euler forces at the inner and outer radii. Furthermore, adding obstacles in the angular direction also passively promotes or inhibits mixing by configuration. This quantitative investigation provides valuable information for designing and developing high throughput and multiplexed point-of-care LOCDs.

## 1. Introduction

Microfluidic systems are being utilized in many applications, such as point-of-care diagnosis, material synthesis, and biomedical analysis, due to tremendous advantages, such as efficient and faster reactions, cost-effective processes, easy and precise control, and high throughput [1,2,3,4,5,6]. To employ these systems for a variety of applications, it is necessary to incorporate numerous physicochemical phenomena. The chemical reaction, in particular, is the fundamental function of the microfluidics system. However, the small scale makes mixing and reacting solutions difficult due to the low Reynolds number (*Re*; the ratio of viscous force to inertial force) regime [5,7]. In this regime, advection is minimal and molecular diffusion is the dominant source of mixing, which we parametrize using the Péclet number *Pe* = *UL*/*D*, where *U* is fluid velocity, *L* is characteristic length, and *D* is diffusivity. The Péclet number for microfluidic systems is generally low, but the reaction time can be relatively long, particularly for materials with low diffusivities, such as large-sized DNA, cells, colloidal particles, etc., in a microchannel or chamber [8,9,10]. In order to accomplish high throughput and multiplex processes, the technique of mixing enhancement is required, even when the reaction quantity is extremely small, with ~*O*(*µ*L) to *O*(*f*L). Many researchers have investigated to resolve this problem by increasing the mixing performance using active or passive mixers. Following the wide usage of photo-lithography and soft-lithography for PDMS microfluidic chip manufacturing, these active or passive mixing systems were implanted utilizing the same fabrication techniques, which are nonetheless expensive and time-consuming [11,12,13,14]. Furthermore, incorporating microfluidics with mechanical motion or complicated functions would significantly increase manufacturing costs and time. Therefore, many researchers are using other alternative methods. As high-resolution production technologies are being developed, several researchers are using alternative techniques, such as 3D printers, laser cutters, and injection molding, etc. [15,16,17,18,19].

Active mixers use external energy sources, such as ultrasound energy, thermal energy, magnetic fields, and electrical fields [20,21,22,23,24,25,26,27]. Under applied ultrasound energy, acoustic streaming resulting from dissipated ultrasound energy is effective in mixing because it induces flow agitation. Furthermore, ultrasonication causes the collapse of a microbubble in liquid due to cavitation and creates a microjet and flow streaming, which enhances mixing. Thermal energy has also been shown to be effective for mixing. As the temperature increases, the fluid viscosity reduces, and the molecular diffusion increases. Mahmud et al. [24] explored the effects of ultrasound and thermal energy on the mixing of microfluidics. These two effects were significant for mixing; for 5<Re<100, an ultrasonicated mixer demonstrated a 6–10% improvement in mixing efficiency as the temperature increased from 30 °C to 60 °C. The application of a magnetic field achieves rapid mixing as magnetic beads in the solution subjected to the magnetic field stir fluid [25,26]. Hejazian and Nguyen [27] demonstrated that a non-uniform magnetic field in the Y-channel between deionized (DI) water and ferrofluid led to secondary flow, and higher ferrofluid concentration was more effective in mixing. A magnetic field-driven active mixer could also be applied to purify the water by mixing nanoparticles, which can capture the heavy metal with the water [28,29,30,31]. These active methods enhance the mixing performance rapidly, but need additional force or energy generation devices, as well as complex control and detecting systems. The passive technique, on the other hand, uses diffusion and chaotic advection of fluids to mix fluids by increasing the interfacial area and lowering the diffusion length between solutions [32]. Passive approaches are achieved by complicated channel geometry that generates secondary flow, such as patterns or barriers in the channel, or through diverse channel geometries, such as square-wave-shaped or curved serpentine channels [33,34,35]. Stroock et al. [36] developed the staggered herringbone pattern embedded inside the microchannel. The asymmetric geometry of the herringbone structure caused chaotic flow with the rotation and extension of local flows and the switch of flow centerline, resulting in mixing augmentation. They demonstrated that at an even higher Péclet number of $9\times 10^{5}$, mixing efficiency reached about 90% at a channel distance of 1.7 cm. Other studies on mixing behaviors have been conducted depending on the arrangement, shape, and density of the obstacles [37,38]. The obstacles split and recombined fluids, resulting in enhancing the flow complexity. Through numerical analysis, Alam et al. [33] demonstrated that diamond-shaped obstacles decreased mixing performance compared to circular and hexagonal obstacles for Re<50, and a decrease in the number of obstacles also caused low mixing performance with a decreasing pressure drop. In addition, Babaie et al. [34] numerically analyzed the mixing performance of the serpentine channel with a wavy wall. The curved serpentine channel enhanced mixing with Dean flow, which is induced by a radial pressure gradient. By stretching and folding the fluid, the wavy walls produced additional separation vortices. When *Re* = 100, the serpentine channel with a wavy wall improved the mixing quality by up to 96.24% compared to the basic serpentine channel (71.88%). Without an external source, passive mixers have a relatively simple configuration, but need a relatively long channel length or more complicated geometry to maximize the mixing efficiency with a short channel. 

Researchers have developed the Lab-on-a-CD (LOCD) platform using forces by disk rotation for the fluid motion to reduce the extra power source and system size by getting rid of pumps. The LOCD platform generates centrifugal force by spinning a microfluidic device with a circular shape, and it transports liquid using only a rotational motor rather than a pressure transducer or syringe pump [39]. The LOCD consists of channels and chambers and integrates a series of operations into one device, such as separation, metering, mixing, valving, washing, and filtering [40,41,42,43,44,45,46,47]. For instance, sample solutions and buffers prefilled in each chamber are transferred through the channels and mixed, washed, or filtered in another chamber where a valve regulates fluid flow. This LOCD system is suitable for the rapid diagnosis of point-of-care devices, enabling automation and miniaturization of microfluidics [48,49]. In addition, researchers have recently paid attention to developing self-powered rotating portable LOCDs for on-site experiments using supercapacitors and rubber strings [50]. Due to these advantages, the LOCD platform has been widely used in bio-applications, such as polymerase chain reaction (PCR) amplification, enzymatic, protein-ligand analyses, extracellular vesicle isolation, quantification, and DNA extraction [51,52,53,54,55]. These circular disk-shaped devices can be fabricated by a 3D printer, a laser cutter, or a CNC-milling machine on various polymer substrates [56,57,58], which is even simpler than the conventional lithographic fabrication, without being limited by the circular shape. 

During the rotational motion of a disk-type LOCD, the centrifugal force acts radially outward, which is expressed as $F_{\mathrm{Cen}}=-m\omega \times \left(\omega \times r\right)$ with mass m, rotational velocity vector ω, and radius location vector r, and there are two additional types of force applied: Coriolis force and Euler forces, which are given by $F_{\mathrm{Co}}=-2m\omega \times u$ and $F_{\mathrm{Eu}}=mr\times \left(d\omega /dt\right)$, respectively, with velocity vector u. The Coriolis force is determined by the rotational velocity, which is directed perpendicularly to the velocity of fluid moving in the rotating disk. The Euler force is perpendicular to the centrifugal force applied to the angular direction, and its direction reverses based on acceleration and deceleration. The fluids in the channel and chamber are mixed using these forces, where the channel is a duct where fluid flows with an inlet and outlet, whereas the chamber is an isolated space where fluid is filled without an inlet and outlet. The channel mixer uses rotational velocity to mix and flow the fluid. When the microchannel in the disk spins, the fluid is transported outward from the reservoir by centrifugal force, and solutions are mixed by transverse flow caused by Coriolis force [59]. Several computational and experimental investigations have been conducted to investigate the effects of the rotational velocity, channel structure, and aspect ratio of the channel cross-section on mixing in channel mixers [58,60,61]. Chakraborty et al. [62] analyzed the regime in which Coriolis-based mixing is dominant, based on the magnitude ratio of Coriolis and centrifugal forces, defined as $\beta =\rho \omega d^{2}/8\mu$ dependent on rotation speed ω, where ρ is the fluid density, μ is the viscosity, and d is the hydraulic diameter of the channel. The findings show that mixing was enhanced when β>1 due to the comparatively strong Coriolis effect. The curved channel accelerates mixing even more because both Coriolis and Dean forces work together. For example, near the corner of a rotating U-shaped channel, the direction of the Coriolis force and Dean force is the same or opposite [63]. Therefore, fluids are mixed by continually stirring and flipping them. 

Several investigations have been undertaken to increase the mixing performance of the chamber mixer in a rotational disk device by using additional active or passive approaches, such as magnetic beads or obstacles [64,65,66,67]. Grumann et al. [66] demonstrated that by periodically rotating a disk containing magnetic beads in a chamber over a permanent magnet, the beads experienced an oscillating magnetic field and induced flow advection. Ren and Leung [64] studied the effects of angular acceleration and chamber geometry on mixing efficiency. Another study compared the mixing time until the standard deviation of the concentration falls below 0.1 in the chamber with and without obstacles, and found that mixing takes 17.5 s without obstacles and 8 s with them [67]. Although many researchers are interested in enhancing the mixing efficiency in the chamber mixer, as above, there are still many aspects that have not been investigated in order to fully understand the effects of rotation conditions and geometric factors of the chamber. For instance, more varying rotational conditions can be considered by changing not only the angular acceleration, but also the rotation period. In addition, the chamber shape or the arrangement of the obstacles could be suggested for efficient mixing, based on previous studies that considered the variation of the chamber volume and the insertion of the obstacles. 

In these aspects, the mixing performance of a spinning disk chamber as an active mixer subjected to various rotation conditions and chamber geometries were comprehensively investigated in this study. First of all, the effects of angular acceleration and deceleration were analyzed to increase the mixing performance of the chamber, and to consider the combined effects of Coriolis and Euler forces. In addition, the effect of the chamber aspect ratio at a constant chamber volume was analyzed, and a suitable geometry for efficient mixing was proposed. This chamber shape effect in a confined geometry can be applied to various applications, especially to the fields where only a limited amount of sample can be handled, such as high-value-added bio-samples [68,69]. Moreover, an additional passive approach was used to determine the extent to which the mixing performance was enhanced by introducing obstacles into the chamber, and even the proper arrangement of the obstacles to expect efficient mixing was suggested. Using numerical simulation, we evaluated the impacts of Coriolis and Euler forces imparted to the mixer chamber, rotating the fluid inside while the disk rotates. In addition to the simulation, experiments were carried out with a 3D-printed disk, incorporating a mixing chamber and brushless direct current motor (BLDC) motor system for rotating the disk. Precisely acquired and post-processed experimental results were in good agreement with the simulation results, both qualitatively and quantitatively, so mixing performance could be compared using the mixing index (*MI*). We expect our chamber mixer analysis will be able to suggest the design of the disk chip mixer in a wide range of practical applications, e.g., bio-sensor technology, cell lysis processes, water and food quality analyses, and soil qualification, etc. [68,69,70,71].

## 2. Materials and Methods

The mixing chamber was designed using commercial 3D modeling software (SOLIDWORKS, Dassault Systems, Vélizy–Villacoublay, France). We used COMSOL Multiphysics 5.4 (COMSOL Inc., Burlington, MA, USA) with microfluidics and chemical reaction engineering modules for numerical simulation. Computations were performed using a workstation with a Windows 10 (64-bit) operating system that includes Intel(R) Xeon(R) CPU E5-2697 v4 running at 2.30 GHz and 128 GB RAM. For the experiment, the rotational mixing chamber and the screw valves were fabricated using a 3D printer (Projet MJP2500 plus, 3D systems, Rock Hill, SC, USA) with resins of a clear transport UV-curable polymer (VisiJet M2R-CL) and wax (VisiJet M2 SUP). As shown in Figure 1a, the printed chamber was rotated using a BLDC motor (BLAM4K30A0750; MDROBOT, Goyang, Korea), while the Arduino control system (Arduino Due; Arduino SRL, Torino, Italy) was connected to the motor driver (MDA1K; MDROBOT, Goyang, Korea). The fluid mixing motion was recorded using a high-speed camera (FASTCAM Mini UX100, Photron, San Diego, CA, USA) with a zoom lens (Zoom 6000, Navitar, Rochester, NY, USA) and under a 12 V and 30 W power of LED light (Nanumvision, Seoul, Korea). The dye solution was made by dissolving the methylene blue powder (molecular weight 319.85 g/mol, M4159, Sigma-Aldrich, St. Louis, MO, USA) of 0.1 wt.% in DI water. MATLAB R2019b (MathWorks, Natick, MA, USA) was used for image processing to calculate the mixing performance.

### 2.1. Numerical Approach

The numerical model for geometry and rotating conditions is illustrated in Figure 1b,c. The basic numerical model includes the height (h), inner radius ($r_{i})$, outer radius ($r_{o})$ and the arc length (ℓ) of the mixing chamber, which were 1, 18, 22, and 10 mm, respectively. The angular velocity (ω) was set as a triangular waveform from 0 to 1000 rpm every 3 s for 6 cycles, as shown in Figure 1c. To conduct the parametric study, we ran multiple simulations by varying the aspect ratio of the chamber (ℓ/w), angular velocity, and the number of periods. Deionized (DI) water and the 0.1 wt.% methylene blue solution at 20 °C were used as the liquid for the simulation and experiment. The momentum equation is given as:(1)$$\rho \frac{\partial u}{\partial t}+\rho \left(u\cdot \nabla \right)u=\nabla \cdot \left[-pI+\mu (\nabla u+{\left(\nabla u\right)}^{T}\right]-\rho \omega \times \left(\omega \times r\right)-2\rho \omega \times u+\rho r\times \frac{d\omega }{dt},$$
where ρ is the fluid density (=998 kg/m^3^), p is the pressure (=1 atm), and I is the unit vector. In the conventional Navier–Stokes equation (Equation (1)), the second to fourth terms on the right-hand side are the additional terms, indicating the volume force expressions for centrifugal, Coriolis, and Euler, respectively. Two different liquids (i.e., DI water and a blue-colored solution) are mixed through advection and diffusion. The governing equation for convection-diffusion is given as follows:(2)$$\frac{\partial c_{i}}{\partial t}+u\cdot \nabla c_{i}=D_{i}\nabla^{2}c,$$
where $c_{i}$ is the species concentration and $D_{i}$ is the diffusion coefficient, which in this study was set to $6.74\times 10^{-10}\;m^{2}/s$ for the methylene blue solution [72]. Initial methylene blue solution concentrations were set to 0.1 wt.%, and no-slip boundary conditions were set for all chamber walls. The equations were solved iteratively using the generalized minimal residual (GMRES) solver. The relative tolerance criterion for convergence was set at 0.005. The mixing performance was quantified based on the homogeneity of two solutions and was calculated through the mixing index $MI=1-\sqrt{\overset{n}{\underset{i=1}{\sum }}{\left(c_{i}-\bar{c}\right)}^{2}/n}/\bar{c}$, where $\bar{c}$ is the averaged concentration, and n is the number of nodes. The value of MI would be 0 before mixing and 1 for fully mixed.

To find the optimal number of meshes for simulation accuracy and low computational cost, the grid independency test was performed by varying the number of grid elements from 51,264 to 578,304. The basic chamber model (as shown in Figure 1b) with an aspect ratio of 2.5 was used, and the rotating condition was set as shown in Figure 1c. The variation in the MI at 18 s was small enough when the number of the elements exceeded 400,000, as reported in Table 1. The computation time for trial 6 required approximately 14 h to get 18 s of mixing. The MI values for mesh element numbers of 443,802 (5th trial) and 578,304 (6th trial) were 0.510 and 0.494, respectively. Therefore, we simulated chamber geometries with at least 400,000 elements.

### 2.2. Experimentation

While experimenting, the 3D-printed mixing chamber was rotated in the range of 0 to 3000 rpm, with an accuracy of ±1 rpm, using an in-house-built BLDC motor control system. Arduino sent a pulse–width modulation (PWM)-type voltage signal to the motor driver. The motor driver converts the PWM-type voltage to the rotation of the motor. The 3D-printed chamber was connected to the BLDC motor using a 3D-printed coupling. The chamber geometry was the same as the simulation condition mentioned in the Numerical Approach section. As shown in Figure 1d, the 3D-printed disk chip includes five mixing chambers, and the whole chip diameter and height are 55 mm and 10 mm, respectively. To visualize the mixing, we matched the rotation of the disk using a high-speed camera at 125 frames per second, with a shutter speed of 1/32,000 s to minimize the image distortion at high-angular velocity. To obtain the accurate visualized images, an initial condition with a sharp interface between solutions (i.e., DI water and 0.1 wt.% dye solution) is required, as depicted in Figure 1b. The gentle injection of dye solution was processed to achieve an initial interface that was as sharp as possible. We used capillary force rather than a pipette to make it. We started by filling the chamber with DI water without bubble formation, and then we injected the blue solutions using a Laplace pressure difference between the inlet and outlet, which is $\Delta p=p_{i}-p_{o}\approx 4\gamma \left(1/d_{i}-1/d_{o}\right)$, where γ is the surface tension of the solution, and $d_{i}$ = 2.5 mm and $d_{o}$ = 6 mm are the diameters of the inlet and outlet reservoirs, respectively. To inject the dye solution, DI water was placed with a hemispherical shape at the outlet, then the blue dye solution was placed in the inlet reservoir. The pressure difference generated by the radius of curvature difference is ~0.67 N/m^2^, and the blue dye solution flows gently without any pressure disturbance, which can be easily generated if we use a pipette. As a result, a sharp interface was created as an initial condition prior to the experiment, as shown in Figure 1d. The thickness of the interface was ~1.4 mm for the basic chamber, and this value varied slightly depending on the chamber geometry. Furthermore, 3D-printed screw valves were used to isolate the mixing from the fluid flow through the channels while the disk rotates. It has a flat-ended nail-like shape, with a radius of 1.25 mm and a length of 5 mm, and an inside square channel with a cross-section area of 0.7×0.5 mm^2^. The fluid flows through the channel when this valve channel is aligned with the channel of the disk, but it does not flow when this screw valve is tuned at 90°. After closing this valve, we performed the mixing experiment by rotating the disk, and images were collected using a high-speed camera. When the grey scale of the extracted pixel data from high-speed camera images was normalized from 0 to 1, the mixing performance was estimated using the MATLAB image processing tool. The mixing index *MI* using the intensity is expressed as $MI=1-\sqrt{\overset{n_{p}}{\underset{i=1}{\sum }}{\left(I_{i}-\bar{I}\right)}^{2}/n_{p}}/\bar{I}$, where $I_{I}$ is the measured intensity of each pixel, $\bar{I}$ is the averaged intensity of the chamber area, and $n_{p}$ is the number of pixels. Here, the value of intensity I directly gives the dye concentration after calibration using the previously reported method [73].

## 3. Results and Discussion

### 3.1. Mixing Behavior of Rotating Chamber

When the rotational velocity ω was varied as a triangular waveform from 0 to 1000 rpm every 3 s (Figure 1b), fluid flow was induced, and two solutions were mixed in the chamber by the Coriolis and Euler forces, which are proportional to the rotational velocity (ω) and rotational acceleration (α=dω/dt), respectively. The Euler force is also proportional to the radius, and it increases as the distance between the origin and the chamber increases. The Euler force difference in the radial direction of the disk generates the rotational flow, as shown in Figure 2a, indicated by the red lines. The Coriolis force, on the other hand, causes rotational motion in the cross-sectional plane by generating the parabolic profile variation along the radial direction, as shown in Figure 2b. These velocity differences along height generate the secondary flows and cause vortex cells to enhance mixing, as shown in the streamlines of Figure 2b. Furthermore, the directions of the Coriolis force in acceleration and deceleration were opposite, and the directions of the induced secondary flows were also opposite when the Coriolis force direction changed, and the oscillated mixing was induced.

Figure 2c shows the simulation results of solution mixing and their velocity vectors at the horizontal mid-plane at *h*/2 height. In the acceleration condition, the Euler force acted tangentially in a counter-clockwise rotation direction, resulting in a counter-clockwise rotational flow, but in the deceleration condition, the Euler force acted in the opposite direction, resulting in a clockwise rotational flow. As shown in velocity vectors in Figure 2c, the flow velocity was fastest at 3.0 s following a period over a cycle, and slowest at 1.5 s during a cycle with a maximum rpm of 1000. The Coriolis force caused these velocity variations when the rotational velocity changed. The Euler force was dominant over the Coriolis force at a low rotational velocity of the disk (i.e., 0 rpm) at 3.0 s because the Coriolis force magnitude became zero, whereas the Euler force remained constant with time. However, when disk rotational velocity increased, the Coriolis force became more dominating and large. Consequently, the Coriolis force generated the additional secondary flow (as depicted in Figure 2b), resulting in low fluid velocity at high disk rotational velocity. The average fluid velocities were 3.90 and 6.49 mm/s at 1.5 s with 1000 rpm and 3.0 s with 0 rpm, respectively. Compared to the previous studies which only reported that the Coriolis force generated the secondary flow in the vertical cross-section and focused more on the effects of the Euler force [64,67], our result is a new report focusing on the Coriolis force that lowered the average velocity of the fluid, even though it caused additional secondary flow.

The angular displacement (Δϕ), which is the angle of the liquid interface from the radial direction (see in sets of Figure 2d), indicates the angle between the initial and mixing interfaces of the solutions [36]. As the angular displacement increased, the interface area between solutions became larger and enhanced the diffusion and mixing. This angular displacement coordinated with the rotational velocity of ω, as shown in Figure 2d. In the acceleration condition, the interface moved following the counter-clockwise rotational flow from 0 to 1.5 s, and Δϕ became larger and had a maximum angular displacement value ($\Delta \phi_{max}=253{}^{\circ})$ at 1.5 s. In the deceleration condition from 1.5 to 3 s, the direction of fluid flow was reversed, and Δϕ reduced with the decreasing rotational velocity. 

The simulation results are validated by comparing the time sequence of images and their mixing performances to the experimental results, as shown in Figure 2e,f. During the experiment, the images were captured from the top using a high-speed camera, showing the accumulated intensity along with the chamber height. To compare, we used the average intensity of the numerical simulation along with the chamber height. Figure 2e shows that both experiment and simulation results agree very well. Figure 2f shows the calculated mixing performance for both the simulation and experiment. Some errors between the simulation and experiment might be caused by uncontrollable experimental conditions. For instance, although an ideally sharp initial interface could be created in the numerical simulation, a bit thicker interface than the simulation could not be avoided in the experiment due to the diffusion of the solutions, which is also influenced by ambient temperature (Figure 2e, 0 s). The surface roughness of the 3D-printed disk chip or vibration during disk rotation could also cause the differences between the experiment and simulation. Nevertheless, the overall qualitative trends of the experiments and simulations still agree well. As a result, for the basic rotating condition of the triangular waveform from 0 to 1000 rpm every 3 s for 6 cycles, the mixing is ~51% after 6 cycles at 18 s.

### 3.2. Effect of the Rotational Condition

In order to investigate the mixing performance against rotational conditions, the mixing performance was analyzed by varying the rotational velocity and period, while maintaining a chamber aspect ratio of 2.5. We conducted two different sets of numerical simulations and experiments by varying the amplitude of the triangular cycling waveform while maintaining the period (Case 1), and changing the period and amplitude while maintaining the acceleration and deceleration (Case 2). Here, Case 1 shows what happens when the amplitude changes. As shown in Figure 3a, the maximum rotational velocity ($\omega_{\max})$ varied from 1000 to 3000 rpm while maintaining a period of 3 s. As shown in Figure 3b, when time, *t* = 18 s after 6 cycles, the mixing indices in the simulation for maximum rpm of 1000, 2000, and 3000 rpm are 0.510, 0.864, and 0.968, respectively, indicating that higher rpm results in better mixing. However, the experimental mixing performance is somewhat lower than the simulation for a high rotational velocity of 3000 rpm, which we expect from the high-speed image distortion caused by the light source. Figure 3c represents the concentration and fluid flow vectors for both the simulation and experiment at *t* =1.5 s, where MI = 0.204, 0.358, and 0.436 from the simulation results at different rpms. As the maximum rpm increased, the mixing performance was enhanced, as shown by an increase in maximum angular displacement ($\Delta \phi_{1000\;}=253{}^{\circ},\;\Delta \phi_{2000\;}=285{}^{\circ},$ and $\Delta \phi_{3000\;}=434{}^{\circ})$ and averaged fluid velocity ($u_{1000\;}=4.9\;\mathrm{mm}/s,\;u_{2000\;}=8.1\;\mathrm{mm}/s,$ and $u_{3000\;}=10.7\;\mathrm{mm}/s$). This is because the rapid rotational fluid flow was generated at a high maximum rotational velocity, and both Coriolis and Euler forces increased, resulting in less rotational flow caused by the large Coriolis force, but more rotational flow caused by the large Euler force. Thus, this suggests that the effect of the Euler force is dominant over the Coriolis force, inducing more rotational flow when the maximum rotational velocity is increased at a fixed rotation period.

In addition, we conducted Case 2, as shown in Figure 3d, by varying the rotational period and the $\omega_{\max}$ with the same angular acceleration and deceleration to exclude the effect of the Euler force effect on mixing. The rotational periods were 1, 2, 3, and 6 s, along with the maximum angular velocities of 333, 667, 1000, and 2000 rpm, respectively. Figure 3e demonstrates that increasing the rotational period enhances mixing performance, and the modeling and experiment results agree well. When time = 18 s after 6 cycles, the mixing indices for four different conditions are 0.310, 0.404, 0.510, and 0.732, respectively, indicating that higher rpm results in better mixing. For relatively low rotational velocity, no significant difference between experiment and simulation was observed, as shown in Figure 3b. Figure 3f represents the concentration and fluid flow vectors for both simulation and experiment at 3 s for periods of 2 s with rpm_max_ = 667 and 6 s with rpm_max_ = 2000, respectively, where *MI* = 0.180 and 0.380 for each case from the simulation results. As the period increased with increasing maximum rotational velocity, the mixing performance was enhanced, showing higher maximum angular displacement ($\Delta \phi_{2\;s}=145{}^{\circ}$ and $\Delta \phi_{6\;s}=287{}^{\circ}$), but lower fluid velocity ($u_{2\;s}=5.3\;\mathrm{mm}/s$ and $u_{6\;s}=4.1\;\mathrm{mm}/s$). The higher Coriolis force induced by the higher maximum rotational velocity resulted in a lower rotational fluid flow at the period of 6 s compared to the period of 2 s. However, the longer period resulted in a larger maximum angular displacement and a larger interface area between the solutions, enhancing the diffusive mixing between solutions. As a result, it was observed that, as the period increases, diffusion becomes more dominant by increasing the interfacial area, despite the fact that the Coriolis force hinders rotational flow velocity, while the Euler force remains constant.

### 3.3. Effect of the Chamber Aspect Ratio

In addition, the effect of the chamber aspect ratio (ℓ/w) on mixing performance was investigated to determine the optimal chamber shape for mixing. In addition to the aspect ratio of 2.5, as shown in Figure 2 and Figure 3, simulations and experiments were carried out with aspect ratios of 0.625, 1.25, and 5 with a fixed chamber volume to compare. To observe the chamber geometric effect only, the rotational condition was maintained the same as the basic rotating condition in Figure 1b of the triangular waveform from 0 to 1000 rpm every 3 s for 6 cycles. Figure 4a represents the mixing index over time for both the simulation and experiment with a varying aspect ratio of the chamber. When *t* = 18 s after 6 cycles, the mixing indices were 0.663, 0.758, 0.511, and 0.222 for aspect ratios of 0.625, 1.25, 2.5, and 5, respectively. The mixing performance was best when the aspect ratio was 1.25, and the poorest when it was 5. Figure 4b shows the concentration and fluid flow vectors for both simulation and experiment at 1.5 s, where *MI* = 0.279, 0.314, 0.205, and 0.089 from the simulation results. In the higher aspect ratio case of 5, the fluid rotates less (at the angular displacement of $\Delta \phi_{\max}$ = 95°) and has a small effect on mixing because the Euler force difference between inner and outer radii was low due to the small w ($\Delta F_{\mathrm{Eu}}\propto \Delta r=r_{o}-r_{i}=w$) and the relatively long chamber arc length compared to the width, as shown in Figure 4b. The Euler force difference and fluid velocity were high for the aspect ratio of 1.25 and more mixing was observed with a larger $\Delta \phi_{\max}$ of 435°. In the aspect ratio of below 1 (i.e., ℓ/w = 0.625), although the Euler force difference between inner and outer radii was ~2.8 times greater than in the cases mentioned above, the rotational flow velocity was low because the chamber arc length was too short, and the fluid could not accelerate adequately. Therefore, it resulted in low $\Delta \phi_{\max}$ of 350° and poor mixing performance compared to the aspect ratio of 1.25.

### 3.4. Effect of the Arrangement of the Obstacles

We further investigated the mixing performance by locating five square pillars inside the mixing chamber, as shown in Figure 5a. The mixing performance is improved by adding a passive mixing system to the rotating active mixing chamber in the disk. The pillar dimension was 0.5 (*w*)×0.5 (*w*)×1 (*h*) ${\mathrm{mm}}^{3}$ and the chamber was sliced to reveal the side view of the actual printed pillar. From the side-cut image, it can be verified that the 3D printer resolution is sufficient to make a pillar inside the chamber. The influence of pillar position on the mixing performance was analyzed by varying the distances between the pillars in angular $d_{\theta }$ and radial $d_{r}$ directions. The chamber aspect ratio was maintained at 2.5, and the basic rotational condition was used as the value in Figure 1b. Both simulation and experiment results comparing the mixing performance by visualization are shown in Figure 5b. Figure 5c shows the mixing index obtained from both the experiment and simulation over time when $d_{\theta }$ changed from 1 mm to 2.25 mm at fixed $d_{r}$ of 0 mm. The mixing performance was seen to be enhanced when $d_{\theta }$ increases up to 2 mm. However, increasing $d_{\theta }$ even to 2.5 mm rather diminished the mixing performance. For the more detailed analysis, the mixing index was compared at t=18 s, as can be seen in Figure 5d, and the mixing indices were 0.532, 0.551, 0.567, 0.593, 0.600, and 0.556 when $d_{\theta }$ increased from 1 to 2.25 mm. The average fluid velocity was reduced by ~10% as $d_{\theta }$ increases, while the maximum angular displacement remained almost constant at 252°. The average velocity decreases as $d_{\theta }$ increased up to 2.0 mm, but more fluid flowed through the gap between the pillars. Consequently, the mixing was enhanced by generating a larger interface area between the solutions, shown as an example in Figure 5(b-i,b-ii). However, when $d_{\theta }=2.25\;\mathrm{mm}$, the distance between pillars was significant, and the gap between the pillars and the chamber surface was small. As a result, this small gap prevented fluid flow, resulting in diminished mixing performance.

Next, we varied $d_{r}$ from 0 mm to 3.5 mm at a fixed $d_{\theta }$ of 2.0 mm. Figure 5e shows the mixing index of both the experiment and simulation, whereas Figure 5f shows the mixing index at *t* = 18 s at different $d_{r}$. The mixing indices were observed to be 0.600, 0.597, 0.585, 0.544, 0.509, 0.470, 0.452 and 0.493 with an increase in $d_{r}$ from 0 to 3.5 mm. As $d_{r}$ increased, the mixing performance decreased, and once it exceeded 2.0 mm, it became even poorer than without pillars (MI<0.510). This is because the velocity of rotational fluid flow was comparatively low at the center region of the chamber and relatively high at the inner and outer walls of the chamber. For the extreme case, when $d_{\theta }$:$d_{r}$ = 2.0 mm:3.5 mm, there is no pillar effect on mixing since the pillars meet with inner and other walls, resulting in better mixing compared to the case of 2.0 mm:3.0 mm. The example of the mixing performance comparison varying $d_{r}$ is shown in Figure 5(b-ii,b-iii). Here, the pillars acted as obstacles to rotational fluid flow when $d_{\theta }$:$d_{r}$ = 2.0 mm:3.0 mm. The pillars near the chamber wall slowed the fluid flow ($u_{2:0}=4.6\;\mathrm{mm}/s$ and $u_{2:3}=3.6\;\mathrm{mm}/s$) and reduced the maximum angular displacement ($\Delta \phi_{2:0}=253{}^{\circ}$ and $\Delta \phi_{2:3}=125{}^{\circ}$). Therefore, the pillars should be positioned at the center line of the chamber with $d_{r}=0$ mm to enhance mixing. A previous study had shown that the installation of an obstacle improves the mixing efficiency [67]. However, we found that, depending on the position of the pillars, there were cases in which mixing was even reduced compared to the case without the pillar, suggesting that the mixing chamber design should take the position of the pillars well into account.

## 4. Conclusions

The present study reports the experimental and numerical investigation of the mixing performance of two solutions within a mixing chamber mounted on a rotating disk. The mixing performance is investigated under various conditions that have not yet been reported, such as rotational conditions, chamber aspect ratio at a constant volume, and obstacle arrangement in the chamber. After determining the optimal number of grids using the grid-independent test, a numerical simulation using COMSOL Multiphysics was performed to analyze the fluid flow in the chamber and the mixing performance. In addition to numerical simulation, experiments were also carried out using a 3D-printed rotational disk with a mixing chamber and an in-house BLDC motor control system. To obtain a sharp initial interface between solutions, surface tension-induced Laplace pressure was used to load the solutions. The screw-type valves were used to isolate the chamber from the fluid inlets and outlet. Two solutions started to mix within the chamber as the disk rotated with the triangle waveform of rotational velocity. This mixing was induced by the Euler and Coriolis forces generated during the disk rotation. The fluid is rotated by the Euler force difference induced by the radial length difference of the chamber. On top of this, the secondary flow was generated along the depth direction by the Coriolis force difference. The secondary fluid flow caused by the Coriolis force is perpendicular to the rotational flow motion caused by the Euler force. Therefore, the rotational flow was hindered by the increase of perpendicular flow velocity as the Coriolis force increased.

To assess mixing performance, a parametric study was conducted by varying disk rotation velocity and period, chamber shape with various aspect ratios, and positioning five pillars inside the chamber. When the rotation period was fixed, but the maximum rotational velocity was increased, the Euler force had a dominant effect on mixing by creating a strong rotation flow, despite the fact that both the Euler and Coriolis forces were generated simultaneously during disk rotation. When the maximum rotational velocity and period were varied while maintaining the rate of rotational velocity, a large interfacial area between solutions was generated, which facilitated mixing through diffusion. In order to examine the change in chamber shape, several aspect ratios of the chamber were tested while maintaining their volume constant. When the aspect ratio of the chamber was small, the Euler force difference augmented as the distance between the inner and outer radii increased. Consequently, as compared to the large aspect ratio (i.e., ℓ/w = 5), the strong rotational flow at a small aspect ratio (i.e., ℓ/w = 1.25) enhanced fluid velocity by ~1.9 times and mixing performance by ~54% at *t* = 18 s for 6 cycles. However, when the aspect ratio was at ℓ/w = 0.625, the short arc length did not provide enough length to increase flow velocity; consequently, the average velocity was reduced by ~7.2%, and the mixing performance decreased by ~9.4% compared to those of ℓ/w = 1.25. To improve mixing performance even further, five pillars were positioned by varying the radial distance $d_{r}$ and angular distance $d_{\theta }$ between pillars to add passive mixing in addition to active rotational mixing. The pillar promotes or inhibits fluid motion based on the pillar configuration. When $d_{\theta }$:$d_{r}$ is 2.0 mm:0 mm, the mixing performance improves by ~9%. However, when the radial distance $d_{r}\;$> 0 mm, the mixing performance decreased due to pillars obstructing the fluid flow and reducing the average velocity ($u_{2:3}=3.6\;\mathrm{mm}/s\;\mathrm{and}\;u_{2:0}=4.6\;\mathrm{mm}/s$) and mixing performance ($MI_{2:3}=0.600,\;\mathrm{and}\;MI_{2:0}=0.452$). 

In summary, it is recommended that greater angular acceleration or a longer period is beneficial for better mixing. In addition, a small chamber aspect ratio close to 1 is suggested. The passive mixing pillars should be placed in the low-flow velocity zone in the chamber. These suggestions could be implemented in the LOCD diagnostic platform for improved mixing and reaction performance in chemical and biological applications. In order to use disk chip-based mixers for a wider range of practical applications, further studies are needed to examine the effects of sample conditions, such as density and viscosity, or mixing environment on mixing.

## Figures and Tables

**Figure 1 micromachines-13-01218-f001:**
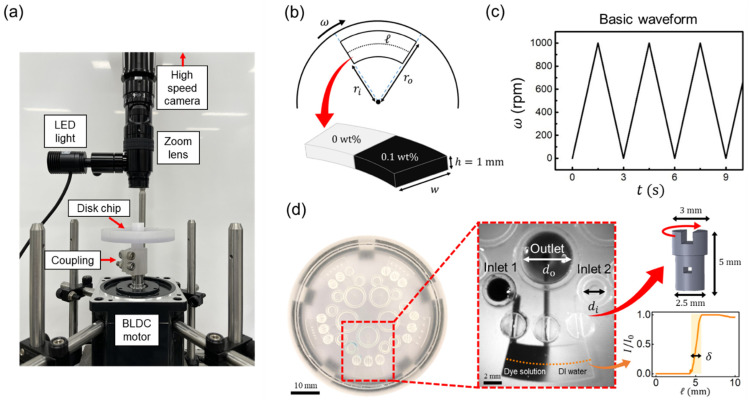
(**a**) Experimental setup; the 3D printed disk is rotated by the BLDC motor. The 3D-printed coupling was used to link the BLDC motor and the 3D-printed disk chip to the motor. A high-speed camera with a zoom lens and an LED light was used to capture the mixing process. (**b**) Schematic of the rotational mixing chamber, where, $r_{i}$ and $r_{o}$ are the inner and outer radii of the chamber, respectively. Here, w is the radial width of the chamber, ℓ is the arc length of the midpoint of the outer and inner radii, and h is the height of the chamber. Initially, the chamber contained a 0.1 wt.% of dye solution and DI water. (**c**) As a basic rotational condition, the rotational velocity (*ω*) and period were set as a triangular waveform with a maximum *ω* of 1000 rpm, a period of 3 s, and a total of 6 cycles. (**d**) The series of rotational type mixers in a disk and the magnification of the chamber mixer. The two different solutions were initially placed in the chamber using inlets 1 and 2, the outlet, and three mechanical valves. $d_{i}$ and $d_{o}$ are inlet and outlet reservoir diameters for gentle injection using capillary pressure. After injecting each solution, the inlets and outlet were closed by rotating the screw valve. Before mixing, the solution showed a sharp interface between the dye and DI water, and the interfacial thickness (δ) was measured to be ~1.4 mm.

**Figure 2 micromachines-13-01218-f002:**
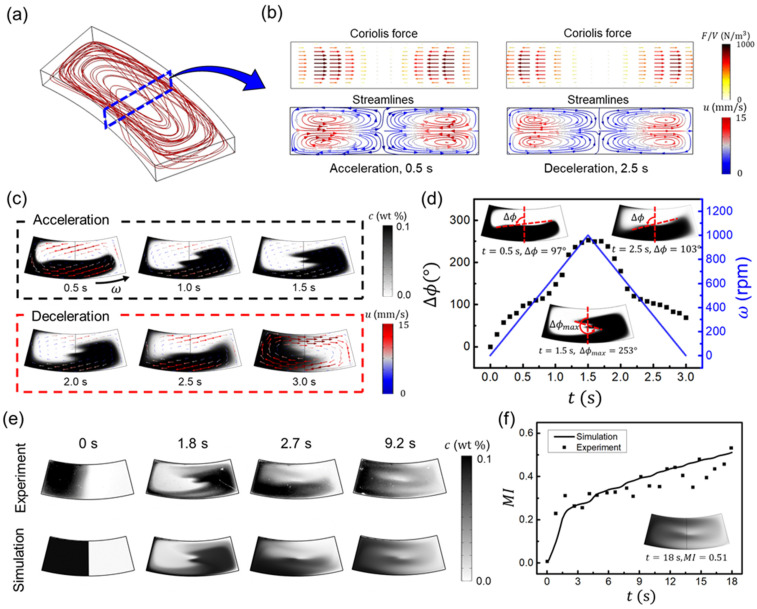
Simulation and experiment results when the rotational velocity ω was varied as a triangular waveform from 0 to 1000 rpm every 3 s. (**a**) Simulation result showing 3D streamlines in the chamber. (**b**) The Coriolis force and the streamlines at 0.5 s and 2.5 s at the cross-sectional mid-plane (blue dashed line in (**a**)). (**c**) Concentration and velocity vectors at the horizontal mid-plane during acceleration and deceleration. (**d**) Angular displacement (Δϕ) and rotational velocity (ω) with time. The term Δϕ indicates the angle, measuring the angular displacement between the initial and leading interfaces of the solutions. (**e**) Comparison of mixing behaviors between the simulation and experiment. (**f**) Mixing index for simulation (solid line), experiment (dots), and the image of simulation at 18 s of mixing where *MI* = 0.51 (inset).

**Figure 3 micromachines-13-01218-f003:**
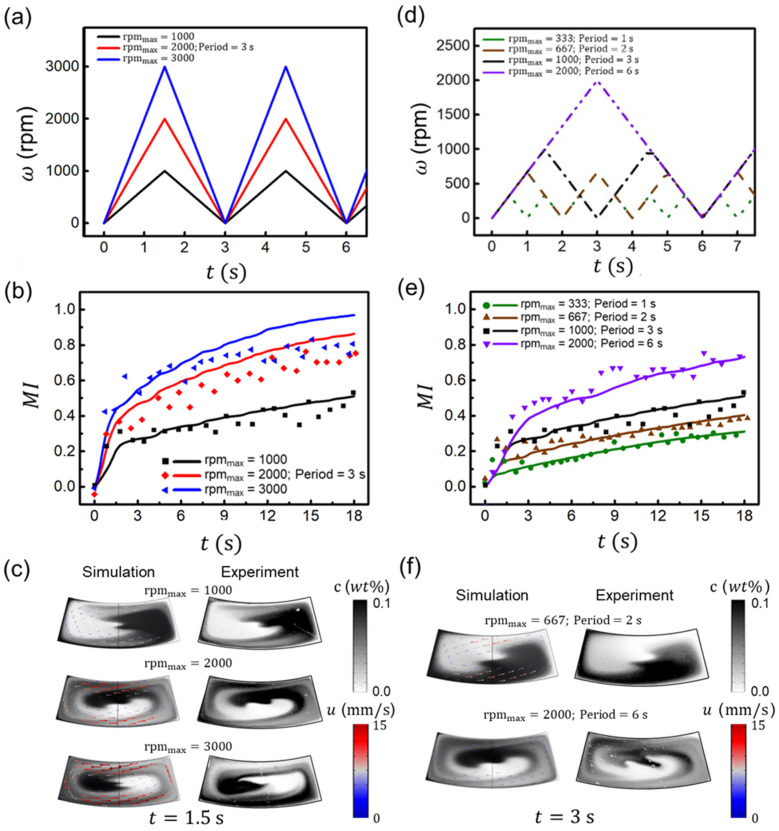
Mixing behaviors with various rotational conditions. Case 1 (**a**–**c**): (**a**) Triangular waveform of rotational velocity of ω for maximum speeds of 1000, 2000, and 3000 rpm with a constant period of 3 s. (**b**) Mixing index for the simulation (solid line) and experiment (dots), with the conditions of (**a**). (**c**) Concentration comparison between the simulation and experiment at 1.5 s for the maximum rpm of 1000, 2000, and 3000 rpm. The arrows in the simulation images depict the velocity vectors. Case 2 (**d**–**f**): (**d**) Triangular waveform for the rotational periods of 1, 2, 3, and 6 s with maximum angular velocities of 333, 667, 1000, and 2000 rpm, respectively. (**e**) Mixing index for the simulation (solid line) and experiment (dots), with the conditions of (**d**). (**f**) Concentration comparison between the simulation and experiment at 3 s for rpm_max_ = 667 with a period of 2 s and rpm_max_ = 2000 with a period of 6 s.

**Figure 4 micromachines-13-01218-f004:**
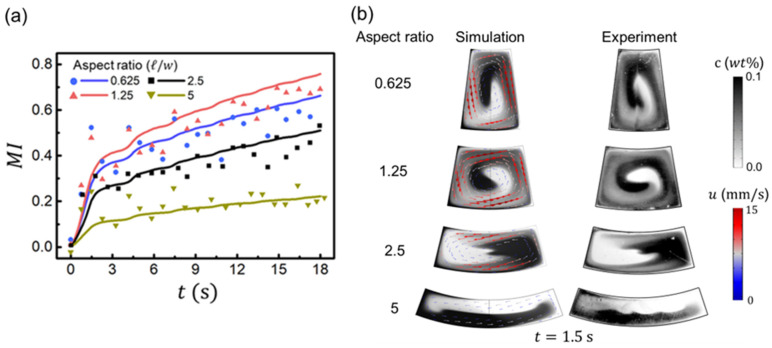
(**a**) Mixing index of the concentration for the simulation (solid line) and experiment (dots), with time for various aspect ratios (ℓ/w) of the chamber. (**b**) Flow visualization of the chamber for both simulation and experiment at t=1.5 s.

**Figure 5 micromachines-13-01218-f005:**
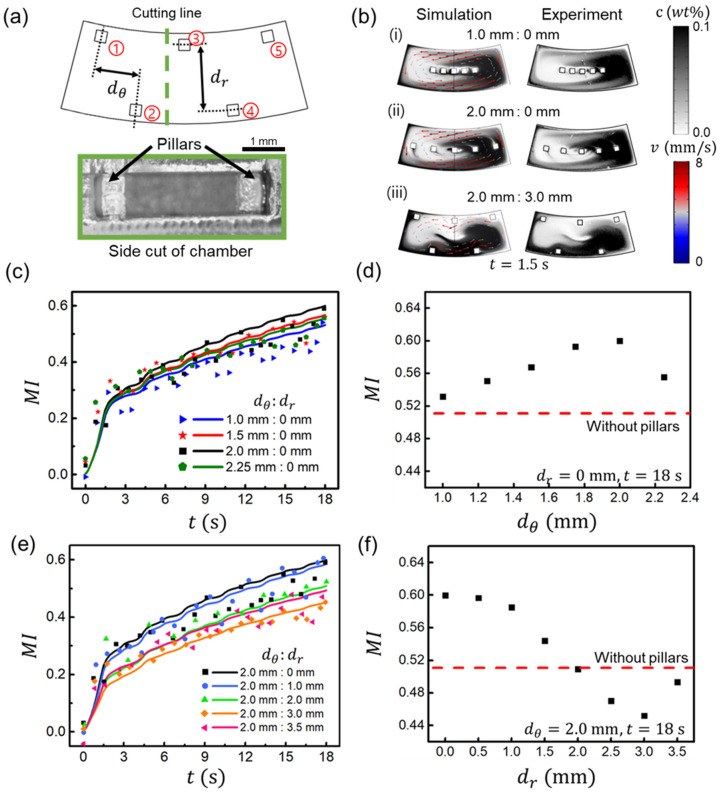
(**a**) Schematic of a rotating mixing chamber with five square pillars of 0.5 (*w*)×0.5 (*w*)×1 (*h*) ${\mathrm{mm}}^{3}$. Here, $d_{\theta }$ and $d_{r}$ are the distances between the pillars in angular and radial directions, respectively. The five pillars from 1 to 5 were fabricated using a 3D printer, as seen in the side-cut image. (**b**) Flow visualization of the chamber for both the simulation and experiment at $d_{\theta }$:$d_{r}$ of (**i**) 1.0 mm:0 mm, (**ii**) 2.0 mm:0 mm, and (**iii**) 2.0 mm:3.0 mm at t = 1.5 s. (**c**) The mixing index for the simulation (solid line) and experiment (dots) with varying $d_{\theta }$ at fixed $d_{r}=0\;\mathrm{mm}$. (**d**) Comparison of the mixing index at *t* = 18 s based on the simulation with and without pillars for different values of $d_{\theta }$. (**e**) The mixing index of the simulation (solid line) and experiment (dots) with varying $d_{r}$ at fixed $d_{\theta }$ of 2.0 mm. (**f**) Comparison of mixing index at *t* = 18 s based on simulation with and without pillars at various values of $d_{r}$.

**Table 1 micromachines-13-01218-t001:** Grid independency test of basic geometry, evaluated using *MI* at *t* = 18 s.

Trial	1	2	3	4	5	6
Mesh element number	51,264	80,208	169,624	288,480	443,802	578,304
Mixing index at 18 s	0.704	0.634	0.582	0.544	0.510	0.494

## Data Availability

Not applicable.

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
