# Peer review of "Enhancing Mixing Performance in a Rotating Disk Mixing Chamber: A Quantitative Investigation of the Effect of Euler and Coriolis Forces"

_micromachines, 2022, doi:10.3390/mi13081218_

Round 1

Reviewer 1 Report

The present manuscript investigates experimentally and numerically the mixing performance in a rotating disk mixing chamber. The manuscript is well-written. On the other hand, the results are not discussed with relevant research articles. Therefore, the manuscript should be reconsidered for publication in Micromachines journal after major revision. The following comments should be taken into account.

1. The originality of the paper needs to be stated clearly. It is of importance to have sufficient results to justify the novelty of a high-quality journal paper. The Introduction should make a compelling case for why the study is useful along with a clear statement of its novelty or originality by providing relevant information and providing answers to basic questions such as: What is already known in the open literature? What is missing (i.e., research gaps)? What needs to be done, why and how? Clear statements of the novelty of the work should also appear briefly in the Abstract and Conclusions sections.

2. Authors should add more explanation about this investigation contribution in the introduction section.  

3. The results are not discussed in the manuscript. Authors should discuss the results and how they can be interpreted in perspective of previous studies and of the working hypotheses. The findings and their implications should be discussed in the broadest context possible and limitations of the work highlighted.  Please include and discuss at least five relevant studies.

4. What is the innovation of this study since there are quite a few other studies under the same topic?

5. The geometry of the experimental device should be explained in detail.

6. Please include the difference between the numerical model and the experimental results at least in the first comparison.

7. In the introduction section, authors may add other methods which are used in the mixing process. The following articles may be used:

a)    Micromixing efficiency of particles in heavy metal removal processes under various inlet conditions, Water (Switzerland), 2019, 11(6), 1135.

b)    A computational tool for the estimation of the optimum gradient magnetic field for the magnetic driving of the spherical particles in the process of cleaning water, Desalination and Water Treatment, 2017, 99, 27–33.

c)   Numerical study of magnetic particles mixing in waste water under an external magnetic field, Journal of Water Supply: Research and Technology - AQUA, 2020, 69(3), 266–275.

d)   Mixing of Fe3O4 nanoparticles under electromagnetic and shear conditions for wastewater treatment applications, Journal of Water Supply: Research and Technology-Aqua, 2022, 71 (6), 671–681.

Author Response

Please see the attached review response file. 

Reviewer 2 Report

The author compared numerical analysis and experiments to improve the mixing performance by configuring the mixing chamber on the rotating disk. This mixing works by causing Euler and Coriolis forces as the disc rotates. In addition, the author variously analyzed the mixing performance through disk rotation velocity, period (time), shape of chamber with various aspect ratios, and obstacle installation. Many analyzes have been compared to numerical simulations and experiments in acceleration and deceleration within a short period of time. Here, I have a few questions.

1. In Figure 2 (a), if the cross section is at both corners rather than the center, how do Coriolis force and streamlines change?

2. In the numerical analysis, when the trials are 6, I am curious about how much time the simulation takes in total. Please tell us the system specifications used for the analysis and the time taken.

3. In the experimentation, how many watts was the LED attached to the lens of the high-speed camera, and was the same intensity maintained in all experiments?

4. After 18 seconds of rotation, what does the mix look like? I am curious about the image after 18 seconds for each experimental condition.

5. How does the mixing index change if the chamber height increases as the aspect ratio of the mixing chamber increases?

6. Analysis was done up to 18 seconds, but the data still doesn't show that the mixing is complete. How long does it take to completely finish mixing (saturation interval)?

In addition to this, we need to add an image to understand the behavior of mixing by 18 seconds.

7. What is the molecular weight of methylene blue powder? Molecular weights are especially important because they affect mixing or diffusion in micro- and nano- scales.

8. In all figures, plots of mixing index versus time were compared to simulated (solid lines) and experimental values (dot symbols). Since some experimental values are mixed with data far from the median, how about expressing the experimental values with line fitting?

Author Response

(The authors gave the same response as above.)

Round 2

Reviewer 1 Report

The manuscript can now be accepted for publication.